# Apparent Absence of BMAL1-Dependent Skeletal Muscle–Kidney Cross Talk in Mice

**DOI:** 10.3390/biom12020261

**Published:** 2022-02-05

**Authors:** Gene Ryan Crislip, Stephanie E. Wohlgemuth, Christopher A. Wolff, Miguel A. Gutierrez-Monreal, Collin M. Douglas, Elnaz Ebrahimi, Kit-Yan Cheng, Sarah H. Masten, Dominique Barral, Andrew J. Bryant, Karyn A. Esser, Michelle L. Gumz

**Affiliations:** 1Department of Physiology and Functional Genomics, College of Medicine, University of Florida, Gainesville, FL 32610, USA; gcrislip@ufl.edu (G.R.C.); cwolff@ufl.edu (C.A.W.); miguel.gutierrez@ufl.edu (M.A.G.-M.); cmdouglas@ufl.edu (C.M.D.); kit-yan.cheng@medicine.ufl.edu (K.-Y.C.); 2Department of Medicine, Division of Nephrology, Hypertension, and Renal Transplantation, College of Medicine, University of Florida, Gainesville, FL 32610, USA; kaesser@ufl.edu; 3Department of Aging and Geriatric Research, College of Medicine, University of Florida, Gainesville, FL 32610, USA; steffiw@ufl.edu; 4Department of Medicine, Division of Pulmonary, Critical Care, and Sleep Medicine, College of Medicine, University of Florida, Gainesville, FL 32610, USA; e.ebrahimi@ufl.edu (E.E.); andrew.bryant@medicine.ufl.edu (A.J.B.); 5Department of Biochemistry and Molecular Biology, College of Medicine, University of Florida, Gainesville, FL 32610, USA; sarahmasten@ufl.edu (S.H.M.); dbarral@ufl.edu (D.B.); 6Myology Institute, University of Florida, Gainesville, FL 32610, USA; 7Center for Integrative Cardiovascular and Metabolic Disease, University of Florida, Gainesville, FL 32610, USA

**Keywords:** circadian clock genes, renal function, integrative physiology

## Abstract

BMAL1 is a core mammalian circadian clock transcription factor responsible for the regulation of the expression of thousands of genes. Previously, male skeletal-muscle-specific BMAL1-inducible-knockout (iMS-BMAL1 KO) mice have been described as a model that exhibits an aging-like phenotype with an altered gait, reduced mobility, muscle weakness, and impaired glucose uptake. Given this aging phenotype and that chronic kidney disease is a disease of aging, the goal of this study was to determine if iMS-BMAL1 KO mice exhibit a renal phenotype. Male iMS-BMAL1 KO and control mice were challenged with a low potassium diet for five days. Both genotypes responded appropriately by conserving urinary potassium. The iMS-BMAL1 KO mice excreted less potassium during the rest phase during the normal diet but there was no genotype difference during the active phase. Next, iMS-BMAL1 KO and control mice were used to compare markers of kidney injury and assess renal function before and after a phase advance protocol. Following phase advance, no differences were detected in renal mitochondrial function in iMS-BMAL1 KO mice compared to control mice. Additionally, the glomerular filtration rate and renal morphology were similar between groups in response to phase advance. Disruption of the clock in skeletal muscle tissue activates inflammatory pathways within the kidney of male mice, and there is evidence of this affecting other organs, such as the lungs. However, there were no signs of renal injury or altered function following clock disruption of skeletal muscle under the conditions tested.

## 1. Introduction

Circadian rhythms are prominent in most physiological processes. The control of these rhythms is mediated in part by a central clock that resides in the superchiasmatic nucleus of the brain, but peripheral circadian clocks also influence rhythms in physiological function [1]. The molecular clock consists of a transcription translation feedback loop. In brief, the core proteins aryl-hydrocarbon-receptor-nuclear-translocator-like protein (ARNTL; or BMAL1) and circadian locomotor output cycles kaput (CLOCK) positively regulate the transcription of the *Period* and *Cryptochrome* genes; the Period (PER) and Cryptochrome (CRY) proteins feedback on and inhibit the activity of BMAL1/CLOCK (reviewed in [2,3]). The disruption of circadian-controlled processes is associated with many pathologies, including cardiovascular and kidney diseases [4,5]. To date, the cross talk between different organ systems following targeted circadian disruption is not well understood. This study focuses on the deletion of the core clock protein BMAL1 in skeletal muscle and the potential ramifications this clock disruption has on the kidney.

The skeletal-muscle-specific BMAL1-inducible-knockout (iMS-BMAL1 KO) mouse model has previously been described to possess characteristics similar to those seen in aging. These mice exhibit an altered gait, reduced mobility, muscle weakness, and impaired glucose uptake [6,7]. Kidney disease is synonymous with aging as the prevalence of kidney diseases increase along with age [8]. Another characteristic reported in the iMS-BMAL1 KO mice is that they display increased tendon calcification [6]. The role of kidneys in the acid–base balance and mineral homeostasis directly links bone health with kidney disease. Consequently, we hypothesized that iMS-BMAL1 KO mice could potentially be a model of kidney disease.

The goal of this study was to determine if disruption of the clock in skeletal muscle leads to renal dysfunction. The deletion of BMAL1 in skeletal muscle causes an increase in inflammatory markers in the kidney. However, neither renal function nor kidney histology appear to be affected by iMS-BMAL1 KO. These data suggest that the consequence of clock disruption in skeletal muscle leads to minimal changes in the kidney under the conditions tested.

## 2. Materials and Methods

Animals. The mouse model used in this study was generated using floxed exon 8 BMAL1 mice [9] crossed with skeletal-muscle-specific, Cre-recombinase mice (human skeletal actin–Cre). Cre was activated by injections of tamoxifen (made in 15% ethanol in sunflower seed oil) 5 weeks prior to studying (2 mg/day; 5 days ip.) to create iMS-BMAL1 KO. Vehicle (15% ethanol in sunflower seed oil)-injected Cre+ mice were used as controls. Male virgin young mice were studied at 17–19 weeks, and aged mice were studied at 13–14 months. All experiments were conducted in accordance with the National Institutes of Health *Guide for the Care and Use of Laboratory Animals* and approved and monitored by the University of Florida Institutional Animal Care and Use Committee and the North Florida/South Georgia Veterans Administration Institutional Animal Care and Use Committee. Mice were housed in temperature- (20–26 °C) and humidity-controlled, 12:12 h light–dark cycled rooms. Mice were provided ad libitum access to water and standard 18% protein rodent chow (no. 2918, Harlan Teklad/Envigo, Madison, WI, USA) unless otherwise noted.

Metabolic Cage Study. A cohort of young male mice (5 control and 5 KS-BMAL1 KO) were maintained in metabolic cages to collect urine and monitor food/water intake [10]. A powder base gel diet (Envigo Teklad Custom Diet) containing 1% agar was prepared and used throughout the metabolic cage collections. Mice were given 3 days to acclimate to the metabolic cages prior to the beginning of collections on a control diet (0.25% NaCl; 0.6% K; Envigo 99131). Following one day of recording with a control diet, mice were then treated with a potassium-deficient diet (0.25% NaCl; 0.0001% K; Envigo 99134) for five days. Urine samples were collected every 12 h at the end of the daylight and nighttime periods (Zeitgeber time 0 and 12). Urine output, food intake, and water intake were recorded. Urine electrolyte concentrations were measured by a flame photometer (Cole-Parmer Model 2655-00, Vernon Hills, IL, USA) according to manufacturer’s instructions. Mice were euthanized, and tissue was collected at Zeitgeber time 6. 

Body Composition. Body composition was quantified in conscious young male mice before and five days after administration of a potassium-depleted diet. Measurements were carried out at Zeitgeber time 4–6 using an EchoMRI Quantitative Magnetic Resonance Body Composition Analyzer (Echo Medical Systems, Houston, TX, USA) [11].

Phase Advance Protocol. A separate cohort of aged male mice were singly housed in cages equipped with a running wheel with ad libitum access to chow (no. 2918, Harlan Teklad) and water. The cages were stored in a light-controlled box with constant air exchange (Actimetrics, Wilmette, IL, USA). The lighting schedule was maintained on a 12:12 h light–dark cycle using a green LED (~200 lux) light source. After one week of baseline, the dark cycle was advanced by 7 h, and the mice were maintained on this new 12:12 h light–dark cycle schedule for one week. This process was repeated a second time to produce a 14-h phase advance. Mice were euthanized at Zeitbeger time 2 in groups of 2–3, and tissue was collected 7–10 days after the second week of phase advance.

Measurement of Glomerular Filtration Rate. Glomerular filtration rate (GFR) was measured in aged male mice before and 6–8 days after the 14-h phase advance protocol using a transdermal monitor (MediBeacon GmbH, St. Louis, MO, USA). Mice were removed from light-controlled boxes but maintained in the dark. Using only red light, transcutaneous measurement of fluorescein isothiocyanate (FITC)-labeled sinistrin was injected via the tail vein to assess GFR at Zeitgeber time 18–21 (7.5 mg sinistrin/100 g body weight made up to 100 µL in saline).

Recombination Specificity. To confirm the specificity of *Bmal1* recombination, we completed PCR using 40 ng of skeletal muscle genomic DNA and primers for the recombined and non-recombined alleles. (fwd primer: ACTGGAAGTAACTTTATCAAACTG, rev primer: CTGACCAACTTGCTAACAATTA, recombination primer: CTCCTAACTTGGTTTTTGTCTGT). The forward and reverse primers for the floxed *Bmal1* allele yield a 431-bp product. The second forward primer 5′-CTCCTAACTTGGTTTTTGTCTGT-3′ was included to detect the recombined product, which shows a band at 572 bp. PCR reaction products were run on a 1.7% agarose gel, and identification of a genomic band and a recombination-specific band confirmed all tamoxifen-treated animals were tissue-specific KO as previously described [7,12].

Immunohistochemistry. Upon anesthetization with inhalant isoflurane, right kidneys were collected and immediately transversely cut in half then stored in periodate-lysine-2% paraformaldehyde for approximately 48 h at 4 °C. Fluid was then changed to PBS to prepare for processing. Kidney samples from each animal were embedded in paraffin, and 4-micrometer-thick sections were cut and mounted on glass slides. Sections were either stained with hematoxylin and eosin or stained with a 3,3′-diaminobenzidine chromogenic substrate (Vector) to target BMAL1 (D2L7G Cell Signaling; 1:3000; RRID: AB_2728705, Danvers, MA, USA). Sections were observed by light microscopy (Nikon E600 equipped with a Nikon DXM1200F digital camera, Melville, New York, NY, USA). Group identifiers were removed from each slide, and BMAL1 expression was examined or slides were scored for tubular injury using the following as criteria: percentage of tubules that showed signs tubular necrosis, a lack of brush border, tubular dilation, and protein cast formation.

Immunoblotting. The cortical regions of the kidneys from young male mice were dissected. Tissue was homogenized using T-PER Tissue Protein Extraction Reagent and a protease–phosphatase inhibitor cocktail (Thermo Scientific, Waltham, MA, USA). Protein concentration for each sample was determined by BCA (Pierce, Thermo Scientific). Protein (25 µg) was separated on a 4–20% Tris–HCL precast gel (Bio-Rad, Hercules, CA, USA) and transferred to a polyvinylidene difluoride membrane. The membrane was stained with Ponceau for 5 min and imaged before washing with Tris-buffered saline (TBS). The membrane was blocked overnight at 4 °C with 5% BSA in TBS plus 0.1% Tween (TBS-T) and then incubated overnight at 4 °C with anti-uncoupling protein 1 (UCP1; U6382 MilliporeSigma; 1:1000; RRID: AB_261838, St. Louis, MO, USA) or anti-GAPDH (14C10 Cell Signaling 2110; RRID: AB_561053). The membrane was washed for 30 min in TBS-T then incubated with horseradish-peroxidase-conjugated anti-rabbit secondary antibody. After an additional wash with TBS-T for 30 min, detection was performed using SignalFire ECL reagent (Cell Signaling Technology) with a 30 s exposure time for imaging. Densitometry was performed using Fiji, and protein abundance was normalized to Ponceau staining.

Multiplex Immunoassays. Cortical and medullary regions were dissected from half a kidney from aged male mice. Tissue was homogenized in either 300 µL (cortex) or 250 µL (medulla) of buffer (1% protease–phosphatase inhibitor cocktail (Thermo Scientific) 1% 1 M ethylenediaminetetraacetic acid, 0.1% Tween 20 in phosphate-buffered saline). Protein concentration for each sample was determined by BCA (Pierce, Thermo Scientific). Homogenates were measured in duplicate for selected analytes using commercial multiplex immunoassay kits (cat#’s MCYTOMAG-70K, MBNMAG-41K, and MKI1MAG-94K; EMD MilliporeSigma) on a MILLIPLEX^®^ Analyzer 3.1 xPONENT System (Luminex 200, Austin, TX, USA) with data analysis via MILLIPLEX Analyst software. The average intra-assay CVs were <5%, and the average inter-assay CVs were <15%. The data were converted from pg/mL to pg/mg, using the previously determined protein concentrations.

Renal Mitochondria Isolation and Respirometry. Fresh renal cortical tissue was dissected from mice following phase advance. Mitochondria were isolated using the Mitochondria Isolation Kit for Tissue (#ab110169; Abcam, Waltham, MA, USA) following the manufacturer’s instructions. Protein concentration of the mitochondrial preparation was determined using a BCA Protein Assay Kit (Pierce, Thermo Scientific). Mitochondrial respiration of the isolate was determined in duplicate in a previously calibrated oxygraph chamber maintained at 37 °C (Oroboros O2K; Oroboros Instruments, Innsbruck, Austria) containing respiration buffer (MiR05; 0.5 mM EGTA, 3 mM MgCl_2_·6H_2_O, 60 mM lactobionic acid, 20 mM taurine, 10 mM KH_2_PO_4_, 20 mM HEPES, 110 mM D-sucrose, and 1 g/L BSA essentially fat acid-free, pH 7.1) [13]. Oxygen concentration of the respiration buffer was kept between air saturation (~200 µM) and 50 µM. Oxygen consumption rate (OCR; ρmol O_2_/s/mg mitochondrial protein) was measured using the following substrate-uncoupler-inhibitor-titration (SUIT) protocol (concentration of reagents noted in parenthesis are final within chambers): (1) LEAK (L) respiration was assessed after TCA cycle stimulation with NADH-linked substrates pyruvate (5 mM), malate (2 mM), and glutamate (10 mM) to support electron flow through complex I (CI) of the electron transport system (ETS; E); (2) oxidative phosphorylation (OXPHOS; P) was stimulated with adenosine diphosphate (ADP; 2.5 mM) and recorded as P_CI_; (3) addition of succinate (10 mM) supported convergent electron flow through complexes I and II of the ETS (P_CI+II_); (4) quality of the mitochondrial preparation was assessed through testing mitochondrial outer membrane integrity by adding cytochrome c (10 µM); (5) the uncoupler carbonyl cyanide 4-(trifluoromethoxy) phenylhydrazone (FCCP; 0.5 µL-steps of a 0.1 mM stock solution) was titrated step-wise until maximum uncoupled respiration was reached and recorded as maximum ETS capacity (E_CI+II_); (6) addition of rotenone (0.5 µM) inhibited complex I of the ETS, and the remaining OCR was recorded as maximum ETS capacity supported by complex II substrate (E_CII_); (7) addition of antimycin A (2.5 µM) inhibited complex III of the ETS and thereby all electron transport to complex IV, and the remaining OCR was recorded as residual, non-mitochondrial oxygen consumption (ROX) and subtracted from all preceding OCRs; (8) oxygen consumption was then stimulated again with the addition of N,N,N′,N′-Tetramethyl-p-phenylenediamine dihydrochloride (TMPD; 0.5 mM; in the presence of ascorbate (2 mM) to avoid uncontrolled autoxidation of TMPD) as an artificial substrate for reducing cytochrome c; (9) the chemical background oxidation rate in the presence of TMPD/ ascorbate was assesed by adding the complex IV inhibitor sodium azide (100mM), and was subtracted from the preceding flux, resulting in maximal capacity of cytochrome c oxidase activity (E_CIV_). Oxygen consumption data were acquired and analyzed using DatLab vs. 7.4 (Oroboros Instruments). Flux control ratios (FCR) were calculated as oxygen consumption at any given respiratory state relative to maximal ETS capacity (E_CI+II_).

Flow cytometry. Lung tissue was digested, and flow cytometric analysis was performed as previously described [14].

Statistical analyses. Graphical data are presented as mean ± SEM. Two-way ANOVA, with repeated measures when possible, was used to analyze differences between four groups. Sidak’s multiple comparisons test was used to compare mouse groups within treatment groups. Student’s *t*-test was used to compare two groups. Analysis was performed using GraphPad Prism version 9.2 software (GraphPad Software Inc., San Diego, CA, USA). Statistical significance was defined as *p* < 0.05.

## 3. Results

### 3.1. Verification of the iMS-BMAL1 KO Model

The knockout for the mouse model in this study was induced by tamoxifen; therefore, the effectiveness of this injection was assessed for all mice. Recombination PCR for BMAL1 on male skeletal muscle tissue from iMS-BMAL1 KO yielded two bands to confirm recombination by tamoxifen injection, one 431 bp band that represents the floxed *Bmal1* allele and a 572 bp band that represents the recombination product [12] (Figure 1A). A single 431 bp band indicates lack of recombination as seen in vehicle-treated mice. Knockout was verified in all mice via recombination PCR following euthanasia. In this mouse model, BMAL1 should only be knocked out within skeletal muscle. BMAL1 protein expression was assessed in kidney sections to ensure there was no effect. Analysis of immune-labeled BMAL1 (brown stain) demonstrated that BMAL1 expression in the kidney is unaffected in iMS-BMAL1 KO mice (Figure 1B). Brown BMAL1 nuclear staining was found in all renal cells in both groups.

### 3.2. Fluid and Solute Handling under Basal Conditions and Following Potassium Depletion

Because the kidney is so resilient to physiological perturbation, especially in C57Bl/6 mice [15], we challenged a cohort of mice using dietary potassium deprivation. We have previously shown that kidney-specific BMAL1 male KO mice exhibit a sodium handling phenotype in response to a zero K diet [10]. Both control and iMS-BMAL1 KO mice excreted 34% less sodium following a potassium-depleted diet compared to baseline; however, there was no difference in sodium excretion between the two groups from urine collected during the active or inactive periods (Figure 2A,B). As expected, potassium excretion dropped to minimal levels within 24 h following potassium depletion during the active and inactive periods (Figure 2C,D). There was no difference between control and iMS-BMAL1 KO mice in potassium excretion. Total body water increased by 6% following potassium depletion in both groups (Figure 2E). Additionally, there was a trend for iMS-BMAL1 KO mice to have more total body water than control with iMS-BMAL1 KO mice with 2–3% higher levels than controls, although this was not significant (*p* = 0.06).

### 3.3. Mitochondrial Function Assessment

Previously, metabolic inefficiencies have been shown to be linked to BMAL1 deletion [7,12,16,17]. To determine if an effect of skeletal muscle BMAL1 knockout influenced renal mitochondria, we first evaluated the expression of the mitochondrial uncoupling protein 1 (UCP1) in the renal cortex of iMS-BMAL1 KO mice compared to control. Basal renal cortical UCP1 protein expression tended to be higher in the iMS-BMAL1 KO mice compared to controls but did not reach significance (Figure 3A,B; *p* = 0.058). Due to the subtle differences in cortical UCP1, a more rigorous approach to assess the effect of iMS-BMAL1 KO on renal mitochondrial function was carried out by performing high-resolution respirometry assays on isolated mitochondria from cortical regions of kidneys. Renal cortical mitochondrial respiratory function in any respiratory state measured (L, P_CI_, P_CI+II_, E_CI+II_, E_CII_, E_CIV_) as well as flux control ratios were similar between the groups (Figure 3C,D). Respiratory control ratio, an indicator of coupled respiration (P_CI_/L), averaged 3.4 and 3.3 for control and iMS-BMAL1 KO mice, respectively. Spare (or reserve) capacity, the difference between maximal OXPHOS (P_CI+II_) and maximal ETS capacity (E_CI+II_), was 22% and 19% of E_CI+II_ for controls and iMS-BMAL1 KO mice, respectively, and did not significantly differ.

### 3.4. Inflammatory Markers in the Kidney

Skeletal muscle has previously been demonstrated to cause an increase in inflammation in the kidney by the release of metabolic wastes. Panels of inflammatory and injury markers were measured in the renal cortex and medulla of iMS-BMAL1 KO and control mice. Cortical interleukin (IL) 6 levels were over 7 times greater in iMS-BMAL1 KO mice compared to controls (Table 1). Similarly, there was a trend for IL-6 levels to be higher in the medullary tissue of iMS-BMAL1 KO mice versus controls, but this did not reach significance (*p* = 0.07). The medullary tissue inhibitor of metalloproteinase 1 (TIMP-1) levels was also found to be higher in iMS-BMAL1 KO mice, at nearly 2 times that of controls (Table 1). There was no difference in fibroblast growth factor 23, IL-10, renin, or kidney injury molecule levels in cortical or medullary tissue from iMS-BMAL1 KO versus control mice. Additionally, no difference was found in cortical IL-5, IL-17, or TIMP-1 levels between the two groups (Table 1).

### 3.5. Renal Injury and Function

Behavioral circadian stress, such as jet lag, leads to deleterious effects on overall health [18,19]. Since the dietary potassium deprivation did not appear to differentially affect the iMS-BMAL1 KO mice in terms of renal excretory function, we hypothesized that behavioral circadian disruption might alter renal function in iMS-BMAL1 KO compared to control mice. Phase advance is a common method for causing circadian disruption [20,21]. Mice underwent a 2-week, 14-h phase advance protocol to shift their night/day time periods. GFR was similar between iMS-BMAL1 KO and control mice before and after the phase advance (Figure 4A). There was minimal tubular injury observed in iMS-BMAL1 KO mice and controls following the histological assessment of kidney sections collected after the phase advance (Figure 4B).

### 3.6. Extra-Renal Effects of iMS-BMAL1 KO

Given the lack of a phenotype in the kidney of iMS-BMAL1 KO mice, we tested the lung to determine if skeletal-muscle-specific loss of BMAL1 adversely affects other tissue. Lungs were collected from mice following the phase advance protocol for flow cytometry to assess inflammatory markers [14]. We found that bone-marrow-derived cells’ fitting profile of immunomodulatory monocytic and neutrophilic myeloid-derived suppressor cell (Mo-MDSC and PMN-MDSC, respectively) infiltration was increased in the lungs of iMS-BMAL1 KO mice versus controls (Figure 5). These findings suggest potential reprogramming of the immune cell population with disruption in BMAL1 signaling in skeletal muscle.

## 4. Discussion

The major finding from this study is that BMAL1 deletion in skeletal muscle did not contribute to changes in kidney function or lead to visible kidney injury under the conditions tested. There are various ways by which skeletal muscle can negatively affect kidney function, including the release of metabolic wastes and contributions to insulin resistance [22,23]. Furthermore, disruption of the clock in skeletal muscle produces an aging phenotype in mice (another characteristic of kidney disease development) [6,16]. Despite iMS-BMAL1 KO mice exhibiting characteristics of aging and signs of increased renal inflammation, there was no renal injury or decrease in renal function detected.

Age-induced renal pathological changes include the development of glomerulosclerosis, interstitial fibrosis, and tubular loss [24]. These progressions in kidney injury lead to a decline in GFR and dysfunction in solute and water handling. In this study, GFR and solute handling were assessed to determine kidney function. Mice were challenged with (1) a change in dietary potassium to alter solute handling or (2) a change in the night/day schedule to expose any potential abnormalities in kidney function. Studies in mice and humans consistently show that dietary potassium depletion leads to an increase in sodium retention [25,26,27,28]. Both groups of mice in this study demonstrated sodium retention following a potassium-depleted diet, as expected, but there was no difference in solute handling between the groups before or after dietary manipulation. GFR was also similar between the groups before and after phase advance. We did not observe any signs of renal injury. Importantly, there was no reported difference in feeding or activity between these groups of mice from previous studies [7,12]. Overall, there were no renal function changes similar to what is seen in an aged phenotype.

There is a high abundance of mitochondria in the kidney associated with high oxygen consumption, which is second only to the heart [29]. A supply of ATP is extremely important for renal function, and mitochondrial impairment has been linked to the progression of kidney disease [22,30]. Based on previous reports on metabolic inefficiencies in skeletal-muscle-specific and global BMAL1-KO mice [7,12,16,17], we evaluated renal mitochondrial function. The absence of BMAL1 in skeletal muscle and the associated changes in this mouse model did not affect mitochondrial respiratory function in the kidney.

Electrolyte homeostasis is heavily influenced by the actions of skeletal muscle, particularly in response to potassium deprivation [31]. The potential role of skeletal muscle BMAL1 in the intracellular shift of potassium balance in response to potassium depletion was not assessed here. This study focused on the kidney, which is the final regulator of electrolyte balance. Interestingly, iMS-BMAL1 KO mice exhibited lower potassium excretion during the rest phase under a normal diet compared to control mice. It should be noted that there was no difference seen during the active phase, which is when the majority of excretion takes place. Additionally, when the active and inactive phase data are combined, there is no difference in 24 h excretion. A limitation of this study is that we were unable to assess serum potassium at the end of the dietary potassium depletion study. This is an important parameter to consider when evaluating cross talk between skeletal muscle and kidney. 

The iMS-BMAL1 KO mice exhibited many characteristics of an unhealthy and aged phenotype, including an altered gait, reduced mobility, muscle weakness, and impaired glucose uptake [6,7,12,16]. Indicators for a negative impact of BMAL1 deletion in skeletal muscle on other organs were apparent. This study examined the effect on the kidney. Although elevated levels of TIMP-1 and IL-6 have been associated with the development of kidney disease [32,33], data in this study indicate that kidney function is not affected despite a significant increase in these markers. As demonstrated by an increase in monocyte and neutrophil infiltration of lung tissue from iMS-BMAL1 KO mice, additional organs including the lungs may demonstrate altered function.

Tamoxifen administration has been demonstrated to cause adverse side effects. Tamoxifen has been shown to induce hernia development in male mice [34]. On the other hand, tamoxifen has demonstrated anti-fibrotic characteristics in a kidney disease model [35]. In this study, low-dose tamoxifen treatment was needed for only five days to sufficiently induce knockout of BMAL1 from skeletal muscle. The mice were not maintained on tamoxifen for a long period of time. Additionally, experiments were not conducted on mice until five weeks post-tamoxifen treatment. It is unlikely that tamoxifen affected results for this study.

## 5. Conclusions

In conclusion, male mice with skeletal muscle BMAL1 deletion did not exhibit altered renal function or kidney injury under the conditions tested. The finding of inflammation in the lungs indicates that extra-renal organs may be affected by clock disruption in skeletal muscle, independent of any changes in the kidney. Overall, the kidney exhibits resilience to the aging phenotype exhibited by iMS-BMAL1 KO mice.

## Figures and Tables

**Figure 1 biomolecules-12-00261-f001:**
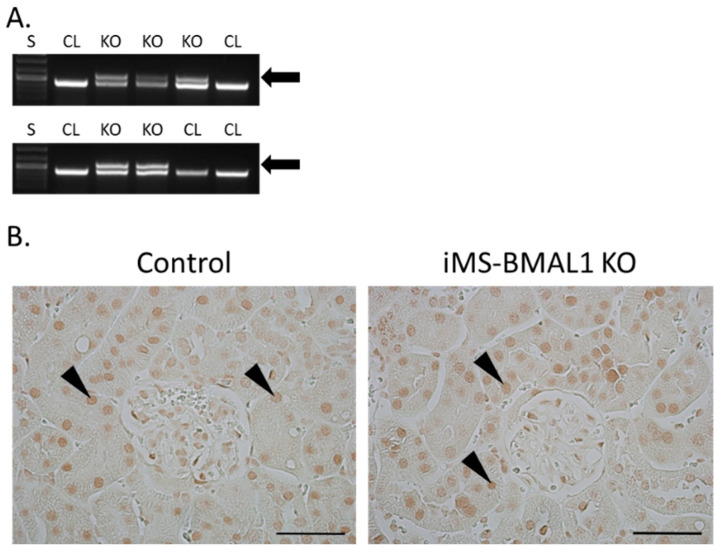
BMAL1 expression is not altered in the kidneys of iMS-BMAL1 KO. (**A**) Recombination PCR assay in skeletal muscle tissue. The forward and reverse primers for the floxed *Bmal1* allele yielded a 431 bp band, which was seen in iMS-BMAL1 KO and control mice. A second forward primer detected the recombined product with a 572 bp band, which was only seen in iMS-BMAL1 KO. Arrow indicates band found in knockout mice. (**B**) Representative images of kidney sections demonstrating BMAL1 protein expression in control and iMS-BMAL1 KO mice. Figure shows the renal cortex. Markers indicate positively BMAL1-stained cells. Scale bar represents 0.05 mm. Mice shown in figure are from the phase advance cohort. S = standards ladder; CL = control mice; KO = iMS-BMAL1 KO mice.

**Figure 2 biomolecules-12-00261-f002:**
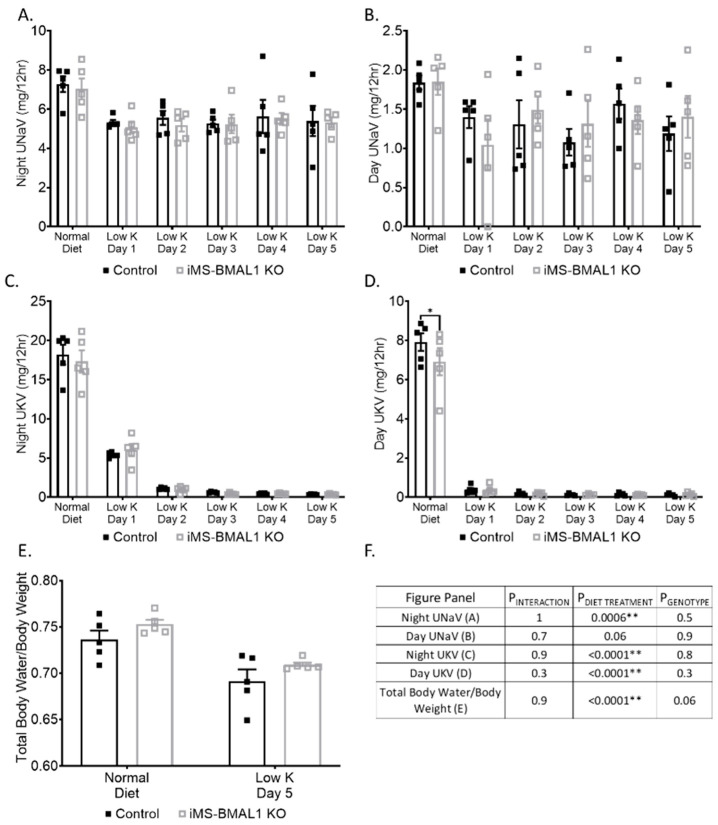
There was no difference in sodium handling between groups following potassium depletion. Sodium excretion during the (**A**) nighttime and (**B**) daytime from male control and iMS-BMAL1 KO mice treated with normal and five days of a potassium-depleted diet. Potassium excretion during the (**C**) nighttime and (**D**) daytime in the same mice. Urine collections were carried out every 12 h. (**E**) Total body water normalized to body weight from mice before and after the five-day treatment with a potassium-depleted diet. (**F**) Statistical analysis for each panel in Figure 2. n = 4–5. Values are mean ± SEM. Two-way ANOVA with was used to compare groups. Sidak’s multiple comparisons test was used to compare between control and KO within treatment groups. * = *p* < 0.05, ** = *p* < 0.001; UNaV = urinary sodium excretion; UKV = urinary potassium excretion; Normal = normal diet; Low K = day of potassium-depleted diet treatment.

**Figure 3 biomolecules-12-00261-f003:**
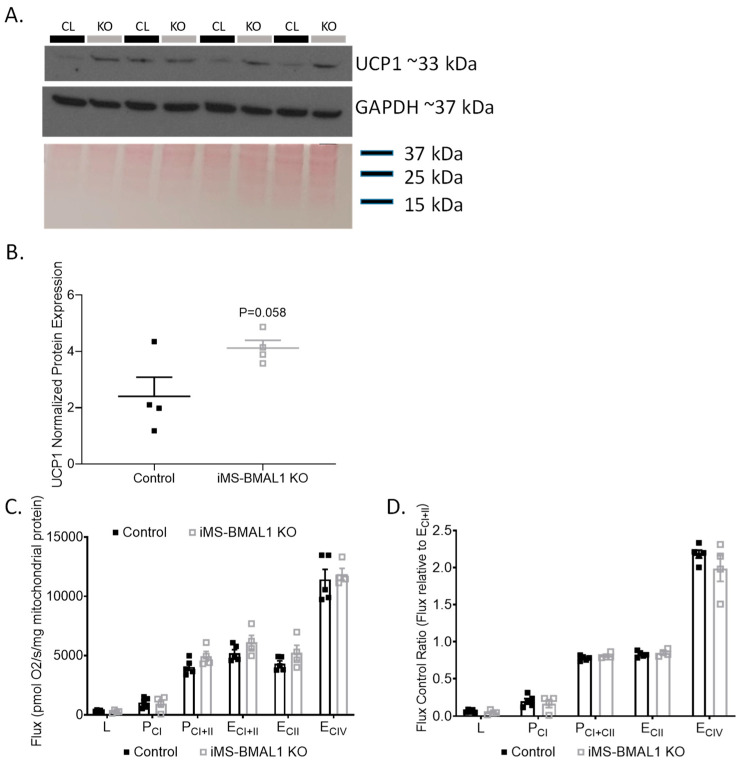
There was no change in renal mitochondrial function in mice with BMAL1 disrupted in skeletal muscle. (**A**) Western blot analysis of basal UCP1 (top panel) and GAPDH (middle panel) protein expression in renal cortical tissue from control and iMS-BMAL1 KO mice. (**B**) Densitometry analysis of uncoupling protein 1 immunoblot normalized to total protein from Ponceau staining (bottom panel in (**A**)). (**C**) Oxygen consumption rates with carbohydrate substrates only from mitochondria isolated from renal cortical tissue from control and iMS-BMAL1 KO mice. (**D**) The ratio of oxygen consumption rates normalized to the maximum (E_CI+II_). n = 4–5. All data are presented as mean ± SEM. Student’s *t*-test was used to compare control vs. KO. CL = control mice; KO = iMS-BMAL1 KO mice; UCP1 = uncoupling protein 1; L = LEAK; P = oxidative phosphorylation; CI = complex I; CII = complex II; E = electron transport system; CIV = complex IV.

**Figure 4 biomolecules-12-00261-f004:**
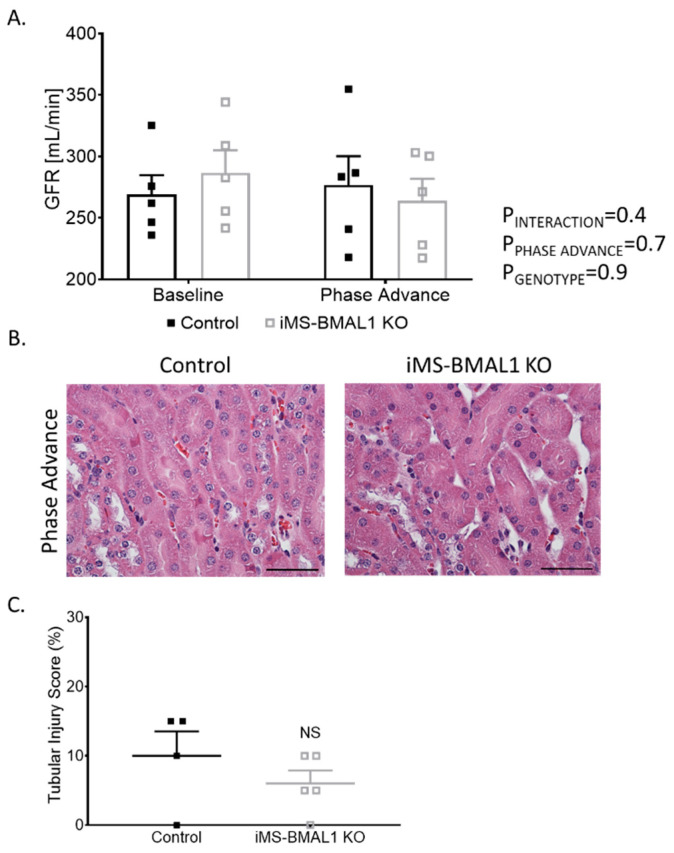
Clock disruption in skeletal muscle does not affect renal function or cause tubular injury. (**A**) Glomerular filtration rate measured via a subcutaneous monitor before and after phase advance in control and iMS-BMAL1 KO mice. n = 5. (**B**) Representative images of kidney sections stained with hematoxylin and eosin from control and iMS-BMAL1 KO mice following phase advance. Cortical region is shown. Scale bar represents 0.05 mm. (**C**) Percentage of tubules that exhibit signs of injury seen in stained kidney sections from both groups of mice. n = 4–5. All data are presented as mean ± SEM. Two-way ANOVA was used to compare groups’ glomerular filtration rate data. Student’s *t*-test was used to compare groups from tubular injury assessment data. GFR = glomerular filtration rate; NS = not significant.

**Figure 5 biomolecules-12-00261-f005:**
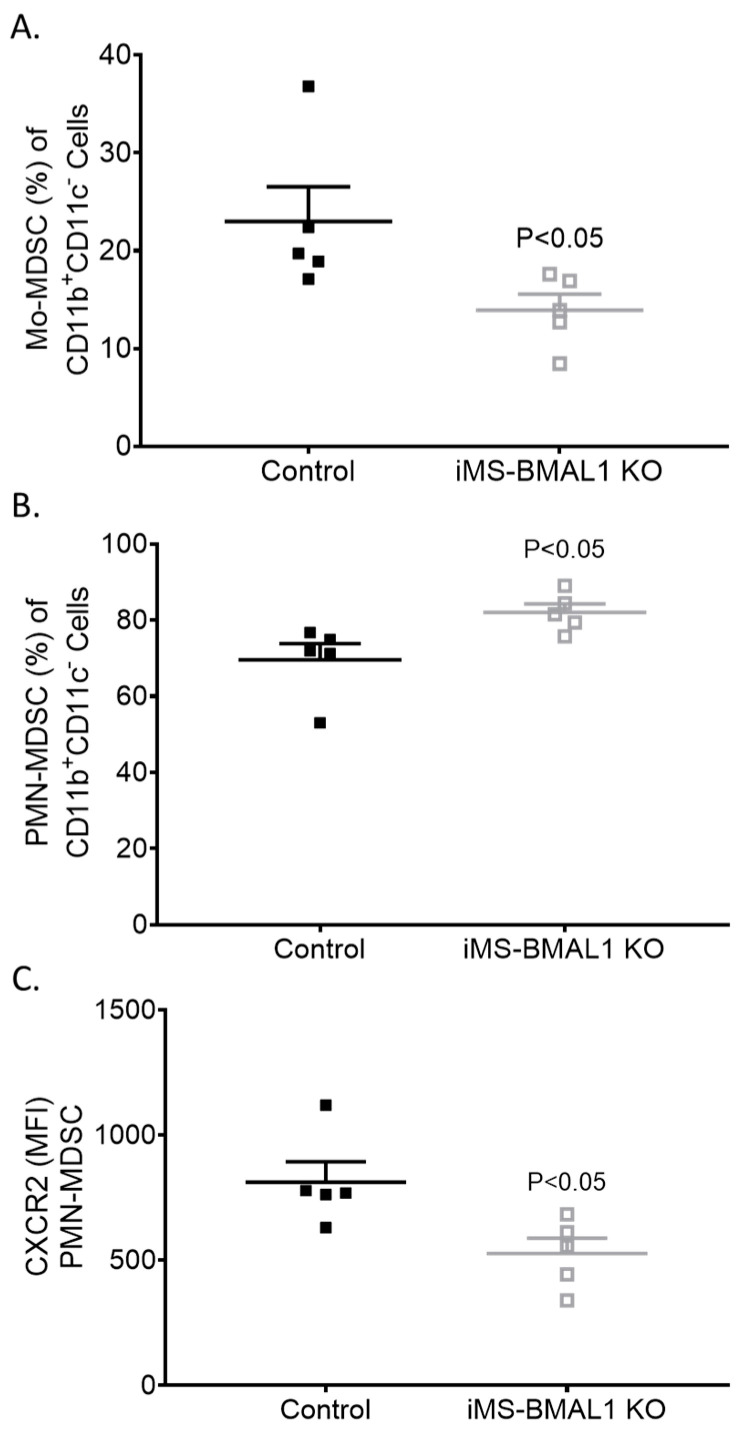
Lungs from iMS-BMAL1 KO had increased immunomodulatory immune cell infiltration. (**A**) Percentage of monocytic and (**B**) polymorphonuclear myeloid-derived suppressor cells in lung tissue from control and iMS-BMAL1 KO mice assessed via flow cytometry. (**C**) Chemokine receptor CXCR2 expression was determined per PMN-MDSC population. n = 5. All data are presented as mean ± SEM. The Student’s *t*-test was used for comparisons between groups. Mo-MDSC = monocytic myeloid-derived suppressor cells; PMN-MDSC = polymorphonuclear myeloid-derived suppressor cells; CXCR2 = CXC motif chemokine receptor 2; MFI = mean fluorescence intensity.

**Table 1 biomolecules-12-00261-t001:** Analytes measured via multiplex immunoassays in renal cortical and medullary tissue from control and iMS-BMAL1 KO mice following phase advance. Values are mean ± SEM with (number of animals per group). Student’s *t*-test.

Analyte (pg/mg)	Kidney Region	Control	iMS-BMAL1 KO
Interleukin 6	Cortex	1.8 ± 0.9 (4)	13.2 ± 4.0 (4) *
	Medulla	1.7 ± 0.1 (3)	6.2 ± 1.7 (4)
Fibroblast Growth Factor 23	Cortex	7.0 ± 1.2 (4)	6.2 ± 0.4 (5)
	Medulla	5.6 ± 3.4 (4)	7.4 ± 1.3 (5)
Interleukin 5	Cortex	0.8 ± 0.1 (4)	0.7 ± 0.2 (5)
Interleukin 10	Cortex	2.4 ± 0.3 (4)	1.9 ± 0.2 (5)
	Medulla	3.3 ± 0.4 (4)	3.6 ± 0.6 (5)
Interleukin 17	Cortex	0.3 ± 0.08 (4)	0.3 ± 0.04 (5)
Renin	Cortex	2710 ± 465 (4)	2620 ± 134 (5)
	Medulla	3082 ± 233 (4)	3937 ± 626 (5)
Kidney Injury Molecule 1	Cortex	112 ± 5.3 (4)	115 ± 11.1 (5)
	Medulla	251 ± 43 (4)	282 ± 29 (5)
Tissue Inhibitor of Metalloproteinase 1	Cortex	122 ± 14 (4)	193 ± 30 (5)
	Medulla	42 ± 5 (4)	82 ± 12 (5) *

* *p* < 0.05 vs. control of same kidney region.

## Data Availability

Data and protocols are available upon request to michelle.gumz@medicine.ufl.edu.

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
