# Peer review of "Apparent Absence of BMAL1-Dependent Skeletal Muscle–Kidney Cross Talk in Mice"

_biomolecules, 2022, doi:10.3390/biom12020261_

Round 1

Reviewer 1 Report

The manuscript, entitled “Impact of BMAL1 Deletion in Skeletal Muscle on the Kidney”, although the male skeletal muscle-specific BMAL1 inducible knockout (iMS-BMAL1 KO) mice have been shown aging phenotype and chronic kidney disease is a disease of aging, the goal of this study was to determine if iMS-BMAL1 KO mice exhibit a renal phenotype. The authors demonstrated that expression of the mitochondrial uncoupling protein 1 (UCP1) in the kidney, glomerular filtration rate, and renal morphology were no significant differences between iMS-BMAL1 KO mice and control mice, suggesting that there are no signs of renal injury or altered function in iMS-BMAL1 KO mice.

This study reveals that male inducible skeletal muscle-specific BMAL1 knockout mice did not represent chronic kidney disease and results are well organized. However, minor issues suggested for attention include:

Minor issue:

  1. Modify the title: the study only focuses on mouse kidney.
  2. Reorganize conclusion in abstract part.
  3. Genotyping of iMS-BMAL1 KO mice: give more information about recombined or non-recombined allele by PCR products in method (Recombination Specificity), result 3.1 and figure legend of Fig1.
  4. The Fig3A WB figure was not clear, authors should provide a better WB represent figure and add normalized house-keeping gene protein as well.

Author Response

Response to Reviewer 2

We wish to thank the reviewer for their timely and helpful review of our manuscript which has helped us improve the paper. Below we address the minor issues raised by Reviewer 2.

Minor issue:

  1. Modify the title: the study only focuses on mouse kidney.

Response: The title has been changed to “Impact of BMAL1 Deletion in Skeletal Muscle on the Kidney in Mice.”

  1. Reorganize conclusion in abstract part.

Response: This part of the abstract has been edited as suggested.

  1. Genotyping of iMS-BMAL1 KO mice: give more information about recombined or non-recombined allele by PCR products in method (Recombination Specificity), result 3.1 and figure legend of Fig1.

Response: These details have been added to the methods as well as the results and figure legend.

  1. The Fig3A WB figure was not clear, authors should provide a better WB represent figure and add normalized house-keeping gene protein as well.

Response: We have added a panel showing the Western blot for GAPDH as well as the Ponceau staining. We believe providing these additional controls helps to clarify the UCP1 results.

Reviewer 2 Report

The manuscript by Crislip et al. explore potential aging-like interaction between the skeletal muscle and renal function. Using the previously described mice model of the specific deletion of circadian clock transcription factor BMAL1 gene in the skeletal muscle, the authors evaluate changes in kidney function, inflammatory pathway, mitochondria properties, and adaptation to the changes in dietary potassium. Overall, despite the interesting idea of the aging-dependent multi-organ cross-talk, the data was primarily negative, indicating minimum or no impact of the skeletal muscle on renal function.

In my opinion, the most intriguing point of the study lies in the potassium balance, which is known to be in control of both the kidneys and skeletal muscle, with potassium channels in both organs being critical molecular determinants of this process. The potassium deficiency increases rates of chronic kidney disease, and the existence of skeletal muscle-kidney signaling axes linked to plasma K+ is an understudied topic in physiology. The authors need to discuss this point better and include these results in the abstract.

Overall, this is a well-written manuscript with rigorous methodology and data analyses. However, there are several points described above needs to be clarified. In addition, I have some minor comments below which should help to improve the manuscript:

-           The title “Impact of BMAL1 Deletion in Skeletal Muscle on the Kidney 2 in Mice” should be changed and reflect the absence in BMAL1 -dependent skeletal muscle- kidney cross talk.

-           If possible, please include blood electrolytes for the potassium depletion experiment.

-           Figure 4 C, y-axes should be adjusted similarly to Figure 5A  

Author Response

We thank the reviewer and Editors for their helpful comments. Below we respond to each issue raised during the review. 

  • Related to the urinary potassium data and the connection between skeletal muscle and kidney during potassium depletion, the reviewer noted "The authors need to discuss this point better and include these results in the abstract."
  • Response: Thank you for this helpful comment, which prompted us to dig a little deeper into the literature. We came across what is now Ref 31, a paper by McFarlin et al., on the topic of coordinate adaptations of skeletal muscle and kidney during dietary potassium depletion. We added the urinary potassium data to the abstract and we have added a new paragraph to the discussion to note that the lack of serum potassium data is a limitation of this work. We also cited the McFarlin data to highlight the important connection between skeletal muscle and kidney for potassium balance. 
  •           The title “Impact of BMAL1 Deletion in Skeletal Muscle on the Kidney 2 in Mice” should be changed and reflect the absence in BMAL1 -dependent skeletal muscle- kidney cross talk.
  • Response: The title has been changed to "Apparent Absence of BMAL1-Dependent Skeletal Muscle-Kidney Cross Talk in Mice."
  •           If possible, please include blood electrolytes for the potassium depletion experiment.
  • We regret that we are unable to provide these data. Following the low K experiment, the mice were transferred back to our collaborator, Dr. Esser. Due to the time at which the animals were later euthanized, it was not possible to collect blood. 
  •           Figure 4 C, y-axes should be adjusted similarly to Figure 5A  
  • Response: Thank you for this comment, this change has been made.